# Development and Validation of a Non-Targeted Screening Method for Most Psychoactive, Analgesic, Anaesthetic, Anti-Diabetic, Anti-Coagulant and Anti-Hypertensive Drugs in Human Whole Blood and Plasma Using High-Resolution Mass Spectrometry

**DOI:** 10.3390/ph16010076

**Published:** 2023-01-04

**Authors:** Catherine Feliu, Celine Konecki, Yoann Cazaubon, Laurent Binet, Damien Vautier, Aurélie Fouley, Claire Gozalo, Zoubir Djerada

**Affiliations:** 1Department of Pharmacology, EA 3801, SFR CAP-Santé, Reims University Hospital, 51 Rue Cognacq-Jay, CEDEX, 51095 Reims, France; 2Institute Desbrest of Epidemiology and Public Health, INSERM, Montpellier University, Department of Pharmacology, Montpellier University Hospital, Avenue du Doyen Gaston Giraud, 34090 Montpellier, France

**Keywords:** high-resolution mass spectrometry, Orbitrap, drug screening, poisoning, plasma, whole blood

## Abstract

(1) Background: In toxicological laboratories, various screening methods can be used to identify compounds involved in intoxication. High-resolution mass spectrometry has been increasingly used in this context for the last years, because of its sensitivity and reliability. Here, we present the development and validation of a screening method that uses liquid chromatography coupled with a high-resolution mass spectrometer. (2) Methods: This method required only 100 µL of whole blood or plasma sample. Pretreatment consisted of a rapid and simple deproteinisation with methanol/acetonitrile and zinc sulphate. This new assay was validated according to international guidelines. (3) Results: To perform the method validation, 53 compounds were selected. The selection criteria were as follows: various chemical structures and therapeutic families (>15), large m/z distribution, positive or negative ionisation mode, and various elution times. The assays showed high selectivity and specificity, with optimal process efficiency. The identification limits, determined using predefined criteria, were established at sub-therapeutic or therapeutic concentrations. Applicability was evaluated using spiked plasma controls and external quality controls. (4) Conclusions: The new method was then successfully applied to routine clinical and forensic samples.

## 1. Introduction

In toxicology laboratories, various qualitative screening methods can be performed to identify compounds involved in poisoning or to monitor compliance. Immunoassays are often used as a primary test. Although these methods are rapid, they are limited in the number of targeted xenobiotics and most of them propose only a drug class diagnosis. Due to lack of sensitivity and specificity, these assays must be confirmed with more appropriate technologies such as mass spectrometry [1,2]. For many years, gas chromatography coupled with mass spectrometry (GC–MS) was considered as the gold standard assay for general screening in toxicology. More recently, the use of liquid chromatography coupled with mass spectrometry (LC–MS) or tandem mass spectrometry (LC–MS/MS) has become more popular due to a reduced analysis time and enhanced sensitivity and specificity. LC–MS is used in various analytical fields, including forensic or clinical toxicology, therapeutic drug monitoring, and clinical pharmacokinetic studies [3,4,5,6,7,8,9]. Multiple procedures have been developed for screening. The first methods used low-resolution mass spectrometry, with triple quadrupole or ion trap technology [3,10,11,12,13,14,15,16]. Analysis was performed according to different approaches, such as targeted screening using multiple reaction monitoring (MRM), or non-targeted screening [3,10,11,12,13,14,15,16]. In recent years, the use of high-resolution mass spectrometry (HRMS) has emerged in toxicology laboratories [17,18,19,20,21,22,23,24,25,26,27,28,29,30,31,32,33]. This technology allows the determination of compounds according to their accurate mass. Previously published papers have described screening methods with a high-resolution Orbitrap mass spectrometer [30,31,32]. The aim of this work was to optimise and validate a screening method in whole blood and plasma using liquid chromatography coupled with high-resolution mass spectrometry (LC–HRMS) and data-dependent acquisition (DDA). We proposed an automated data analysis with a reference library including more than 1400 compounds. This allowed us to obtain a powerful screening concept. We validated a method for a large number of compounds: 53 compounds were used for the complete validation. The development of the method took place in three stages. First, the library was refined. The sample pretreatment was also optimised. Then, the validation method was performed using 53 compounds selected according to chemical structures (>15), pharmacological families (>15), large *m*/*z* distribution, positive (majority) or negative ionisation mode, and various elution times [34,35,36,37,38]. The limits of identification were also determined for 179 compounds in both matrices including 22 benzodiazepines and related substances. Finally, this new validated method was successfully applied in routine clinical and forensic toxicology analyses.

## 2. Results and Discussion

### 2.1. Optimisation

#### 2.1.1. Chromatographic Conditions and Mass Spectrometer Parameters

Different chromatographic parameters were assayed and optimised to ensure good elution of the compounds with correct sensitivity. The first step of optimisation was the mass spectrometer parameters. A mixture of a pure solution of the 53 selected compounds in methanol was infused and then injected to optimise ionisation and mass spectrometer parameters. The second optimisation step was the chromatographic conditions with the choice of column, mobile phases, and elution gradient. Acquity BEH HILIC (50 × 2.1 mm, 1.7 µm), Acquity HSS T3 (150 × 2.1 mm, 1.8 µm), Acquity UPLC^®^ HSS C18 1.8 μm 2.1 × 150 mm (Waters Corp., Milford, MA, USA), and Accucore Phenyl Hexyl UPLC 100 × 2.1 mm, 2.6 µm (ThermoFisher Scientific, San Jose, CA, USA) columns were tested using the previous solution. The chosen column was selected on the basis of the optimal shape of chromatographic peaks. The selected column was the Accucore Phenyl Hexyl UPLC column, as was the case for Helfer et al. [30,31]. The assayed mobile phases consisted of water + formic acid 0.1% (*v*/*v*) (mobile phase A) with or without ammonium acetate and acetonitrile + formic acid 0.1% (*v*/*v*) (mobile phase B). Mobile phases with ammonium acetate provided a better signal strength than water + formic acid 0.1% (V/V). After these choices, the gradient was optimised.

#### 2.1.2. Sample Pretreatment

Sample pretreatment was optimised by testing different liquid–liquid extraction procedures. First, an extraction using 100 µL of the sample, 1 mL of diethyl ether, and 20 µL of carbonate buffer (20%, *v*/*v*) was performed. After vortex mixing and centrifugation, the supernatant was evaporated under nitrogen flow at 40 °C. The dry extract was then reconstituted with 200 μL of water/acetonitrile (50%, *v*/*v*) and formic acid 0.1% (*v*/*v*). Analysis of this sample failed to detect some polar compounds, particularly metformin and baclofen. Different protein precipitation techniques were tested using acetonitrile or methanol [31,39]. The use of these solvents alone did not provide a sufficient signal for all the tested compounds. We therefore combined methanol and acetonitrile precipitation to improve analyte recovery. Subsequently, assays were performed with and without zinc sulphate. These tests showed an improvement of the performance for some analytes with zinc sulphate, especially in whole blood.

In other reported methods, Helfer et al. developed their method in plasma by comparing two sample preparations: precipitation with or without on-line consecutive turboflow extraction [31]. Roche et al. also proposed a semi-quantitative screening in three matrices (plasma, whole blood, and urine) [32]. Our current method proposed validation in two matrices: plasma and whole blood. Our method used the smallest sample volume, 100 µL versus 200 µL [32] or 250 µL [31], which may be an advantage when the volume collected is limited, such as in paediatrics. Regarding sample pretreatment, the procedures also differed. For plasma samples, Roche et al. [32] proposed deproteinisation with methanol before on-line purification with a TurboFlow^®^ system. Helfer et al. described methanol/zinc sulphate precipitation with or without in-line purification by a TurboFlow^®^ system [31]. Most recently, Joye et al. proposed a method of drug screening from dried blood spot using HRMS technology [33].

#### 2.1.3. Library

The library initially provided by the supplier contained 1464 compounds. It has been further incremented with compounds whose identification by a screening analysis was of clinical interest. The following compounds were therefore added to the inclusion list after infusion of a pure solution: baclofen; hydroxychloroquine; anticoagulants (apixaban, dabigatran, rivaroxaban, fluindione, tioclomarol, and phenindione); rodenticides (difenacoum, diphenadione, chlorophacinone); and antidiabetic such as sitagliptine, vildagliptine aztreonam, cefepime, cefotaxime, piperacillin, hydroxyalprazolam, isavuconazole, nordosulepine, norquetiapine, norsertraline, sulpiride, vortioxetine, and oxomemazine. For these compounds, compound name, formula, polarity, high-resolution mass, retention time, isotopic distribution, and HRMS spectrum were recorded to the library. Finally, the library consisted of 1489 compounds.

#### 2.1.4. Screening Data Processing

Different plasma samples spiked with the 53 selected compounds at the concentration of 200 µg L^−1^ as well as a blank plasma sample were prepared and analysed (*n* = 6). Different data processing parameters were studied in order to detect all the compounds in the spiked plasma samples and none in the blank sample. Concerning the identification criteria, the first two major criteria tested were the presence of the high-resolution precursor ion mass and the isotopic pattern. The main limitation of this setting was the inability to discriminate isomeric compounds, such as 6-monoacetylmorphine and naloxone, morphine and norcodeine, O-demethylvenlafaxine, and tramadol. To overcome this problem, a DDA acquisition was set up. In DDA mode, in the first step, the mass spectrometer selected the most intense ions; then, in a second step, they were fragmented and analysed. Finally, the major criteria selected were the presence of high-resolution mass precursor ions, the presence of fragment ions, and the match with the full spectrum of the library spectrum. The minor criteria were retention time and isotopic pattern.

Finally, we proposed a method with an analysis time of 15.25 min. Regarding the other reported method, the analysis time was similar to the procedure of Helfer et al. with precipitation only (17 min). The analysis time was longer with the Turboflow system (33.58 min for Roche et al., and 21 min for Helfer et al.).

### 2.2. Method Validation

As in other reported methods, this present study was validated in accordance with the reference guidelines [36,37,38,40]. For their validation, all authors selected numerous compounds from different therapeutic and chemical classes (14 compounds for Roche et al., 36 compounds for Helfer et al., and 53 compounds for our method).

#### 2.2.1. Interference Studies

1.Selectivity

Analysis of 10 blank plasma samples from individual donors did not identify any interfering compound on the chromatograms (data not shown).

2.Specificity

Our method was able to discriminate all pairs of compounds with the same molecular weight. For each analyte, retention times, accurate mass precursor, and mass tolerance range at 5 ppm are listed below: amitriptyline (RT 6.38 min, 278.19033, (278.188939–278.191721)), venlafaxine (RT 5.14 min, 278.21146, (278.2100689–278.2128511)), alprazolam (RT 6.67 min, 309.09015, (309.0886045–309.0916955)), and warfarin (7.23, 309.11214, (309.1105944–309.1136856)), prazepam (7.93, 325.11022, (325.1085944–325.1118456)), and citalopram (5.73, 325.17107, (325.1694441–325.1726959)). For isomeric compounds, Table 1 summarises the retention time (RT), the accurate mass of the precursor ions, and the five most essential fragment ions with their relative abundance. These results demonstrate the ability of our method to discriminate between them.

3.Evaluating Interferences from Stable-Isotope Internal Standards and carry-over

During the interference evaluation, no analyte of interest was detected (data not shown). Thus, for this analytical method, no interference from stable-isotope IS nor contamination was highlighted.

4.Matrix effect

Results for the matrix effect (ME) are presented in Figure 1 and in Appendix A. The results ranged from 0.86 to 2.28 for plasma and from 0.75 to 2.17 for whole blood, except for metformin. A decreased metformin signal was observed with matrix factors of 14.5% and 10.5% for plasma and whole blood, respectively. For metformin, the relative standard deviations of ME were 18.06% and 13.88% for plasma and whole blood, respectively. This demonstrates that the observed signal suppression had an acceptable dispersion for a screening analysis. Moreover, the addition of metformine-D6 as an internal standard was retained in order to overcome the risk of a matrix effect impairing the identification of metformin.

#### 2.2.2. Recovery and Process Efficiency

The recovery (RE) and process efficiency (PE) are described in Figure 2 and in Appendix A. Recovery ranged from 0.91 to 1.16 for plasma and from 0.70 to 1.21 for whole blood. Relative standard deviations were less than 20% for all compounds. Process efficiency ranged between 0.75 and 2.01 in plasma and between 0.66 and 1.84 in whole blood, except for metformin (0.12 and 0.11 for plasma and whole blood, respectively). Relative standard deviations were less than 20% for all the compounds. All of these parameters are acceptable for the screening method.

#### 2.2.3. Limit of Identification

As described in literature, the identification limit (LOI) was determined as the lowest concentration for which the three major criteria (presence of high-resolution mass precursor ions, presence of fragment ions, and match with the library spectrum) are present [17,30]. LOIs were determined, for both matrices, with a large panel of 179 compounds (Appendix A). The LOIs in plasma were compared to the therapeutic concentrations reported in the literature [41,42]. For the most part of analytes, the LOIs were lower than the therapeutic concentrations. The LOIs were within the therapeutic concentrations for amlodipine (LOI 5 µg L^−1^ for a therapeutic concentration range of 3 to 15 µg L^−1^), buprenorphine (LOI 1 µg L^−1^ for a therapeutic concentration range of 0.5 to 10 µg L^−1^), metformin (LOI 100 µg L^−1^ for a therapeutic concentration range of 100 to 1300 µg L^−1^), and morphine (LOI 10 µg L^−1^ for a therapeutic concentration range of 10 to 100 µg L^−1^). For these compounds, the present method will be able to identify these compounds in therapeutic or toxic use. These results demonstrated the relevance of our method in the investigation of intoxication causes.

The identification limits reported in the different published studies have been displayed in Appendix A. For most compounds, the identification limits were similar between the three methods.

#### 2.2.4. Spectra Reproducibility

The reproducibility of mass spectra was tested on spiked plasma and whole blood at two concentration levels: 10-fold LOI and 3-fold LOI. Between-day identification (*n* = 6) was performed. For each compound, the mean and relative standard deviation of the library score were calculated (Appendix A). The mean of the library score ranged from 45.5 a.u. for levetiracetam to 973.8 a.u. for verapamil. Relative standard deviations were less than 20% for all compounds, except for levetiracetam (25.8%) and amlodipine (22.9%). This assay demonstrated acceptable spectra reproducibility for this screening method, confirming its robustness.

### 2.3. Applicability

Our present qualitative screening assay was tested by inter-laboratory testing of four samples (Appendix A). Our method allowed for the detection of all compounds present in the samples without any false positives.

This assay was successfully applied for identification of compounds in routine toxicological analysis. We propose the description of four cases for analysis for which the toxicological screening analyses were performed using the present method. After a clinico-biological discussion and when it was required, quantitative analyses were conducted with specific methods using LC–MS/MS.

Case 1: A 59-year-old woman, hospitalised in psychiatry, was transferred to the intensive care unit for alleged voluntary drug poisoning. The patient was somnolent and had hemodynamic instability associated with bradycardia 40 beats per minute (bpm) and a blood pressure of 60/30. Routine toxicology screening detected acetaminophen, alimemazine, amlodipine, atenolol, caffeine, fluoxetine, furosemide, ketamine, midazolam and alpha-hydroxymidazolam, zopiclone, and demethylzopiclone. Figure 3 illustrates amlodipine identification in the plasma sample. All the criteria were checked: retention time (a), high-resolution mass precursor ion (b), isotopic distribution (c), fragment ions (d), and HR–MS/MS spectrum similar to the reference library spectrum (e).

In this context, quantifications of drugs inducing hemodynamic instability were performed. Quantification analyses revealed a major polyintoxication to amlodipine (701 µg L^−1^, toxic concentrations from 80 µg L^−1^ [42]), atenolol (12,880 µg L^−1^, toxic concentrations from 2000–3000 µg L^−1^ [42]), and zopiclone (547 µg L^−1^, toxic concentrations from 150 µg L^−1^ [41]). The other analytes were within the therapeutic range.

Case 2: A 76-year-old woman was admitted to the emergency room for alleged voluntary drug poisoning. She was found on the floor at her domicile and showed altered vigilance (11/15 on the Glasgow Coma Scale). She had a previous history of depression, a self-inflicted phlebotomy, and an attempt of self-hanging. The biochemical analysis revealed hypokalemia, hypoglycemia, and elevated transaminases. The toxicological screening revealed the presence of numerous drugs: acetaminophen, alprazolam, bromazepam, caffeine, ceftriaxone, codeine, flecainide, fluoxetine and norfluoxetine, laudanosine, milnacipran, morphine and morphine-3-glucuronide, noscapine, omeprazole, tramadol, tianeptine, verapamil and norverapamil, and zolpidem. Toxic concentrations were quantified for bromazepam (3460 µg L^−1^, toxic from 300 µg L^−1^ and inducing coma from 1000 µg L^−1^ [42]), milnacipran (415 µg L^−1^, therapeutic concentrations: 50–110 µg L^−1^ [41]), zolpidem (641 µg L^−1^, toxic from 500 µg L^−1^ [42]), and tramadol (1650 µg L^−1^, toxic from 1000 µg L^−1^ [42]). Tianeptine was found at 5480 µg L^−1^ (therapeutic concentrations: 30–80 µg L^−1^ [41]), explaining the biological signs of liver alterations. Therapeutic concentrations were quantified for other detected drugs.

Case 3: A 54-year-old woman was admitted to the intensive care unit for an unexplained coma associated with metabolic acidosis. The present routine toxicological screening was performed. The compounds detected for this patient were alimemazine, amoxicillin, bromazepam, canrenone, ketamine and norketamine, lorazepam, midazolam, metformin, morphine, omeprazole, venlafaxine, and norvenlafaxine. In the context of the metabolic acidosis and the screening results, metformin intoxication was suspected. It was confirmed by the quantification of metformin at a concentration of 21,300 µg L^−1^ (therapeutic concentrations: 12.5–2500 µg L^−1^ [43]).

Case 4: Another case of alleged voluntary drug poisoning was reported concerning a woman aged 71 years old. She was admitted to the intensive care unit presenting a coma associated with vasoplegia and cardiogenic shock. The present routine toxicological screening allowed for the detection of acetaminophen, atropine, bisoprolol, cetirizine, dobutamine, hydroxyzine, levetiracetam, lidocaine, lormetazepam, nordiazepam, oxazepam and zopiclone, and desmethylzopiclone. In this context, quantifications of numerous drugs were performed. A lethal concentration was quantified for oxazepam (4750 µg L^−1^, lethal from 3000–5000 µg L^−1^ [42]). A toxic concentration was quantified for hydroxyzine (139 µg L^−1^, toxic from 100 µg L^−1^ [42]). Bisoprolol was found within the therapeutic range (71 µg L^−1^, therapeutic concentrations: 10–100 µg L^−1^ [42]). Levetiracetam, lormetazepam, and zopiclone were found above therapeutic range (109 mg L^−1^, 29 µg L^−1^, and 85 µg L^−1^, respectively, therapeutic concentrations: 10–40 mg L^−1^, 2–10 µg L^−1^, and 10–50 µg L^−1^, respectively [41]). Nordiazepam was found under the therapeutic range (19 µg L^−1^, therapeutic concentrations: 20–800 µg L^−1^ [41]).

Case 5: A 57-year-old man was found unconscious at home with a Glasgow score of 3/15. The environment suggested an autolysis attempt by self-medication intoxication. Blood pressure was 182/144 mmHg, heart rate was 65 bpm, and respiratory rate was 18 breaths per minute. Oxygen saturation was 100%, and body temperature was 35 °C. The patient was transferred to the medical intensive care unit for further management. Routine toxicological screening was performed, revealing the use of alprazolam; cetirizine, citalopram and its metabolites, domperidone, hydroxyzine, ondansetron, morphine and codeine, salbutamol, thiopental, and zolpidem. Thiopental was suspected as being responsible for the condition of the patient. Thiopental was then quantified at the concentration of 21.8 mg L^−1^ by a liquid chromatography method coupled with ultraviolet spectrophotometry. At this concentration, thiopental can induce coma and could be fatal [42]. Other detected drugs were quantified in the therapeutic concentrations range.

## 3. Materials and Methods

### 3.1. Chemicals

7-Amino-clonazepam, amiodarone, amlodipine, atenolol, atropine, bisoprolol, bromazepam, bupivacaine, citalopram, chlordiazepoxide, clobazam, clonazepam, clozapine, diazepam, estazolam, flupentixol, haloperidol, hydroxyzine, lamotrigine, lidocaine, loxapine, mianserin, metformin, midazolam, nordiazepam, mitrazepam, o-demethylvenlafaxine, oxazepam, prazepam, propranolol, risperidone, sertraline, tramadol, verapamil, warfarin, zolpidem, and zopiclone were purchased from Sigma-Aldrich (St. Gallen. Louis, MO, USA). Alprazolam, amitryptiline, aripiprazole, baclofen, clotiazepam, desmethylzopiclone, fentanyl, paroxetine, and venlafaxine were purchased from Lipomed (Arlesheim, Switzerland). Glibenclamide, levetiracetam, loprazolam, lorazepam, lormetazepam, norclobazam, oxazepam, and temazepam were purchased from LGC standard (Teddington, Middlesex, UK). Acetazolamide, cyamemazine, nefopam, and tiapride were obtained from Carbosynth (Newbury, Berkshire, UK). Buprenorphine, cocaine, ketamine, MDA, methadone, morphine, morphine-D3, and amphetamine-D5 were purchased from Cerilliant (Round Rock, TX, USA). Metformin-D6 and trazodone-D6 were obtained from Cluzeau info labo (CIL, Sainte-Foy-La-Grande, France). Acetonitrile, methanol, formic acid, and water, all LC–MS hypergrade, were purchased from Biosolve (Dieuze, France). Zinc sulphate and ammonium formate were purchased from Sigma-Aldrich (St. Gallen. Louis, MO, USA). Whole blood and plasma from healthy donors were purchased from the French Blood Bank (“Etablissement Français du Sang”, EFS, Reims, France).

### 3.2. Preparation of Stock Solutions and Working Solutions

The internal standards were prepared as 1 g L^−1^ stock solutions in acetonitrile. A working internal standard solution was then prepared monthly, containing 1 mg L^−1^ lopinavir-D8, metformin-D6, tramadol-D6, trazodone-D6, and 0.1 mg L^−1^ amphetamine-D5 and morphine-D3.

Other compounds were prepared as 1 g L^−1^ stock solutions in methanol and stored at +4 °C. Working solutions were prepared with appropriate serial cascade dilutions in methanol.

### 3.3. Preparation of Quality Control Samples

Quality controls were prepared in plasma after appropriate dilution of the stock solutions to obtain the following final concentrations: 10 µg L^−1^ for amlodipine and haloperidol; 100 µg L^−1^ for atenolol, bupivacaine, cocaine, cyamemazine, glibenclamide, morphine, prazepam, and venlafaxine; 500 µg L^−1^ for metformine; and 1000 µg L^−1^ for furosemide, ketamine, levetiracetam, and warfarin.

### 3.4. Chromatographic and Mass Spectrometric Conditions

#### 3.4.1. Liquid Chromatography

An ultra-performance liquid chromatographic system with an Ultimate 3000 high-pressure pump (ThermoFisher Scientific, San Jose, CA, USA) coupled with Orbitrap QExactive mass spectrometer (ThermoFisher Scientific, San Jose, CA, USA) was used for the development and the validation of the method. Chromatographic separation was performed using an Accucore Phenyl Hexyl UPLC column (100 × 2.1 mm, 2.6 µm, ThermoFisher Scientific, San Jose, CA, USA), maintained at 40°C. Mobile phases consisted in 2 mM ammonium formate, water, and formic acid 0.1% (*v*/*v*) (mobile phase A) and 2 mM ammonium formate, acetonitrile, and formic acid 0.1% (*v*/*v*) (mobile phase B). A programmed mobile-phase gradient was used at a flow rate of 0.5 mL.min^−1^. The gradient was programmed as follows: 0–0.5 min 99% A, 0.5–10 min 99% to 1% A, 10–11.5 min hold 1% A, and 11.5–15.3 min hold 99% A. The time of analysis and acquisition was 15.3 min including equilibration.

#### 3.4.2. High-Resolution Mass Spectrometry

Heated electro-spray ionisation in positive/negative switching ionisation mode was performed with the following settings: (1) sheath gas 45 arbitrary units (a.u.), (2) auxiliary gas 15 a.u., (3) sweep gas flow rate 1 a.u., (4) spray voltage 3.50 kV, (5) ion transfer capillary temperature 300 °C, (6) S-lens RF level 70 V, and (7) heater temperature 350 °C. Mass spectrometry was performed using full-scan data and a subsequent DDA. Runtime acquisition was 0 to 15.25 min. The settings for full scan data acquisition were as follows: (1) scan range of *m*/*z* 70 to 1000; (2) resolution power of 35,000 FWHM (full width at half maximum) for *m*/*z* = 200; (3) automatic gain control (AGC) target of 1.10^6^ a.u., and (4) maximum injection time (IT) of 120 ms. For data-dependent acquisition mode (DDA), an inclusion list containing 1513 compounds was added. High-collisional dissociation (HCD) was performed on the three most intense ions selected from the full scan. Moreover, a dynamic exclusion for 3 s was planned for the most intense ions. DDA settings were as follows: (1) resolution power 17,500 FWHM for *m*/*z* = 200, (2), AGC target of 1E^5^ a.u., (3) maximum IT of 50 ms, (4) isolation window of *m*/*z* 2.0, and (5) collision energy stepped at 17.5, 35 and 52.5 eV.

Mass calibration was performed once a week in positive and negative mode using an external calibration solution (Pierce^®^, ThermoScientific, San Jose, CA, USA) according to the manufacturer’s recommendations.

#### 3.4.3. Screening Data Processing

TraceFinder Forensic 5.1 was used for LC–MS control, library management, acquisition, and processing. Precursor peaks (*m*/*z*) were detected with a minimal ratio signal-to-noise threshold of 10 a.u. and a mass tolerance of 5 parts per million (ppm); retention time, fragment ions, isotopic pattern, and library search were selected for confirmation as follows:-for retention time, the option “ignore if not defined” was selected, and window override was 60 s;-for isotopic pattern, the fit threshold was 70 a.u., with a mass tolerance of 5 ppm and an intensity deviation tolerance of 30%;-for fragment ions, the option “ignore if not defined” was selected, the minimum number of fragments was one, the intensity threshold was 5000 a.u., the product mass tolerance was 10 mmu, and the MS order was MS²;-for general library NIST settings, MS order was MS² and isolation width was used; for the NIST setting, search type selected was MS/MS, and options “ignore precursor”, “use all peak matching”, “reverse search”, and presearch “off” were selected; a probability threshold of 10 a.u, score threshold of 80 a.u., search index (SI) threshold of 500 a.u., and reverse search index (RSI) threshold of 600 a.u. were selected; precursor and product masses tolerances were 5 ppm and 10 ppm, respectively; library score was selected as a passing value type with a passing value of 20 a.u.

#### 3.4.4. Library

The initial library was kindly provided by ThermoFisher Scientific (San Jose, CA, USA). An update of the inclusion list with the new compounds was performed in our laboratory. For each compound, the library mentioned the compound name, formula, polarity, high-resolution mass, retention time, isotopic distribution, and HRMS spectrum.

### 3.5. Sample Preparation

A total of 100 µL of zinc sulphate 5% (*v*/*v*) and 20 µL of the working internal standard solution were added to 100 µL of whole blood or plasma samples. Deproteinisation with 100 µL methanol followed by 200 µL acetonitrile was conducted. After vortexing for 30 s and centrifugation at 10,000× *g* for 5 min, the organic phase was evaporated under nitrogen flow at 40 °C. The dry extract was then recovered by 200 μL of water/acetonitrile (50% *v*/*v*) containing 0.1% formic acid (*v*/*v*). For chromatographic separation, 20 µL was injected. Figure 4 depicts an example of chromatographic elution for a laboratory-made control containing ten compounds (atenolol, ketamine, cocaine, venlafaxine, bupivacaine, haloperidol, cyamemazine, warfarin, glibenclamide, and prazepam).

### 3.6. Validation Method

The method was validated according to international guidelines for forensic qualitative analyses [36,37,38,40]. Fifty-three compounds were selected to validate this qualitative method according to their chemical structures, pharmacological families, broad *m*/*z* distributions between 100 and 700, ability to be detected in positive or negative ionisation mode, and retention times along the chromatogram (Table 1). Validation was performed on plasma and whole blood.

#### 3.6.1. Interference Studies

1.Selectivity

Selectivity or matrix interference was assessed by analysing ten drug-free human plasma/whole blood samples.

2.Specificity

The ability of the method to differentiate between compounds of the same molecular weight or isomeric compounds was considered.

Differentiation of identical molecular weight compounds was tested: amitriptyline: C_20_H_23_N, 277.18305; venlafaxine: C_17_H_27_NO_2_, 277.20418; alprazolam: C_17_H_13_ClN_4_, 308.08287; warfarin: C_19_H_16_O_4_, 308.10486; prazepam: C_19_H_17_ClN_2_O, 324.10294; and citalopram: C_20_H_21_FN_2_O, 324.16379. Differentiation of isomeric compounds was also tested: O-demethylvenlafaxine/tramadol: C_16_H_25_NO_2_, 264.19581; MDEA/MBDB: C_12_H_17_NO_2_, 208.13321; acepromazine/aceprometazine: C_19_H_22_N_2_OS, 327.15256; morphine/norcodeine: C_17_H_19_NO_3_, 286.14377; 6-MAM/naloxone: C_19_H_21_NO_4_, 328.15433; N-demethylclobazam/oxazepam: C_15_H_11_ClN_2_O2, 287.05818.

3.Evaluating Interferences from Stable-Isotope Internal Standards and carryover

The isotopically labelled compounds used as IS may contain the unlabelled analytes as impurities. Interference from stable-isotope IS was evaluated by analysing a blank matrix sample spiked with the internal standards. The corresponding analytes of interest should not be detected. For carry-over, six blank plasma samples and six whole blood samples were spiked with the 53 selected analytes at a final concentration of 10,000 µg L^−1^. Blank plasma extracts were analysed after the spiked samples (*n* = 6).

4.Matrix effect

The matrix effect defined as ionisation suppression or enhancement was evaluated according to the FDA (U.S. Food and Drug Administration) [35] and EMA (European Medicines Agency) [34] guidelines and literature [6,7,36,38]. For each matrix, two different sets of samples were prepared (*n* = 6), and the peak areas of the neat standard (53 compounds) were compared to the matrix samples spiked with neat standards after extraction. This assay was carried out at two concentration levels (50 and 500 µg L^−1^). The matrix effect factor was calculated by comparing the area under the peak of the spiked matrix after extraction and the area under the peak of the neat solution at the same concentration.

#### 3.6.2. Recovery and Process Efficiency

Recovery (RE) and process efficiency (PE) were evaluated according to the FDA [35] and EMA [34] guidelines and literature [6,7,36]. The recovery was assessed by comparing the area under the peak derived from the matrix spiked before extraction and the area under the peak derived from the matrix spiked after extraction. Finally, process efficiency was performed comparing the area under the peak derived from the matrix spiked before extraction and the area under the peak of a pure solution at the same concentration.

#### 3.6.3. Limit of Identification

Whole blood and plasma samples were spiked with the 179 selected compounds at different concentrations according subtherapeutic, therapeutic, and toxic concentrations, providing a large scale of concentrations [41,42] for each compound. Therefore, 10 µL of the working solutions were spiked into 100 µL of plasma or whole blood. Thereafter, the samples were prepared as described above.

Within-run and a between-run analyses were assessed by analysing 6 samples per level. The limit of identification was determined as the lowest concentration at which the substances were identified in all replicates on the basis of the three following criteria [30]: (1) the presence of high-resolution mass precursor ions, (2) the presence of fragment ions, (3) matching with the library spectrum. Retention time and isotopic distribution were considered as minor criteria. These criteria for limit of identification were in accordance with the classification proposed by Broecker et al. [17] and were described in the paper of Helfer et al. [30].

#### 3.6.4. Spectra Reproducibility

The reproducibility of the MS/MS spectra was investigated by analysing the spiked plasma at three levels of concentration [30]. Between-day assays (*n* = 6) were performed. For mass spectra reproducibility evaluation, library score of the TraceFinder search index values was used as quality match factors. For each compound, mean and relative standard deviation of library score were calculated.

#### 3.6.5. Statistics

GraphPad Prism 5.0 (GraphPad Software, San Diego, CA, USA) was used for statistical analysis. Data are described as mean and standard deviation.

## 4. Conclusions

In this work, we described the development and validation of a large-scale HRMS toxicology screening on both whole blood and plasma. The method acquisition used was a full-scan DDA with an inclusion list of more than 1000 compounds. Sample work-up consisted of a straightforward deproteinisation with only 100 µL of plasma or whole blood. Validation was carried out according to international guidelines and the current literature. This new test is now successfully applied to routine clinical and forensic toxicology analyses. In the coming future, further development and validation of this assay will be performed in urine and other matrices. In addition, the data acquisition can be adapted to allow for the detection of new unknown compounds such as synthetic cannabinoids by using Compound Discoverer (ThermoFisher Scientific, San Jose, CA, USA).

## Figures and Tables

**Figure 1 pharmaceuticals-16-00076-f001:**
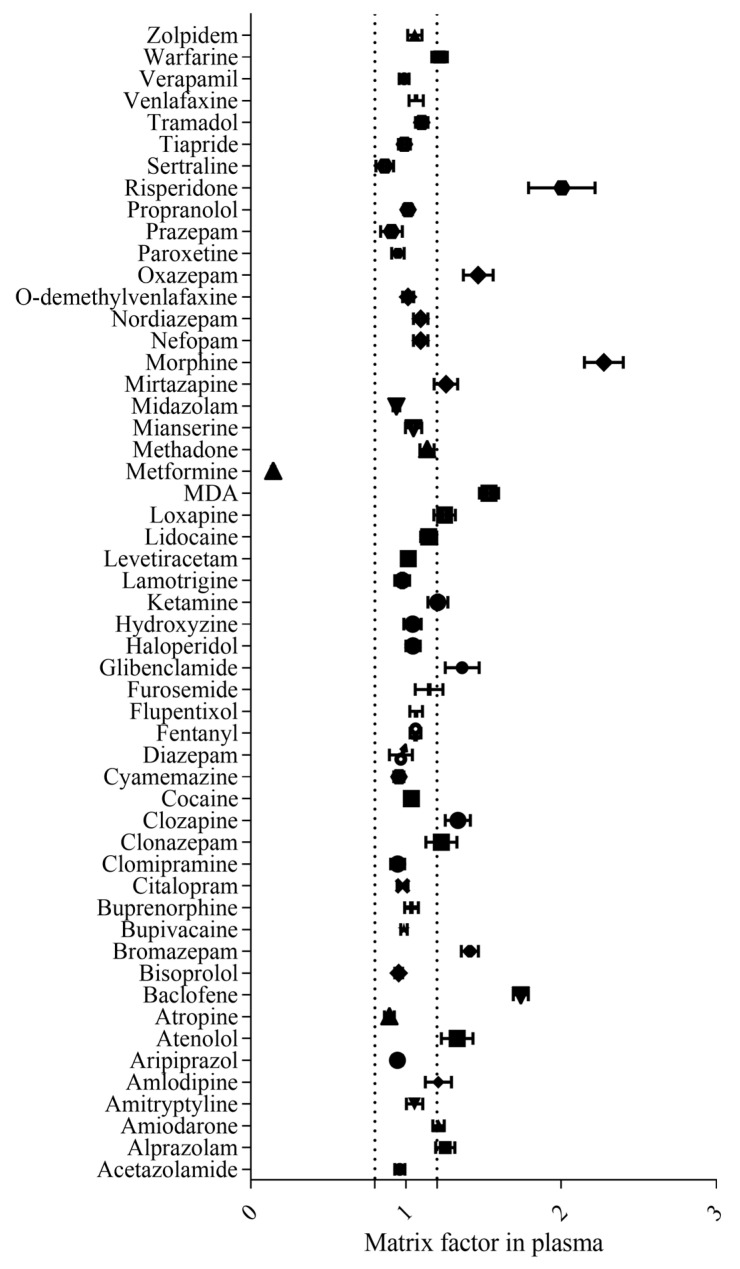
Matrix effect in plasma samples. Matrix effects are represented as mean +/− SEM (*n* = 6) in the *x*-axis. Dashed lines represent +/− 0.2 of the one unit.

**Figure 2 pharmaceuticals-16-00076-f002:**
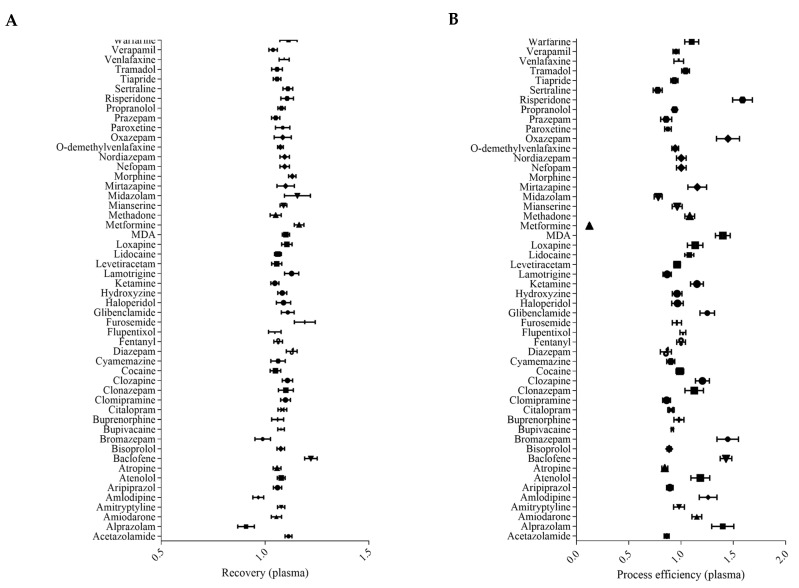
Recovery and process efficiency in plasma samples. Recovery (%) is depicted (**A**) and process efficiency is depicted (**B**). Data are presented as mean +/− SEM for each compound (*n* = 6).

**Figure 3 pharmaceuticals-16-00076-f003:**
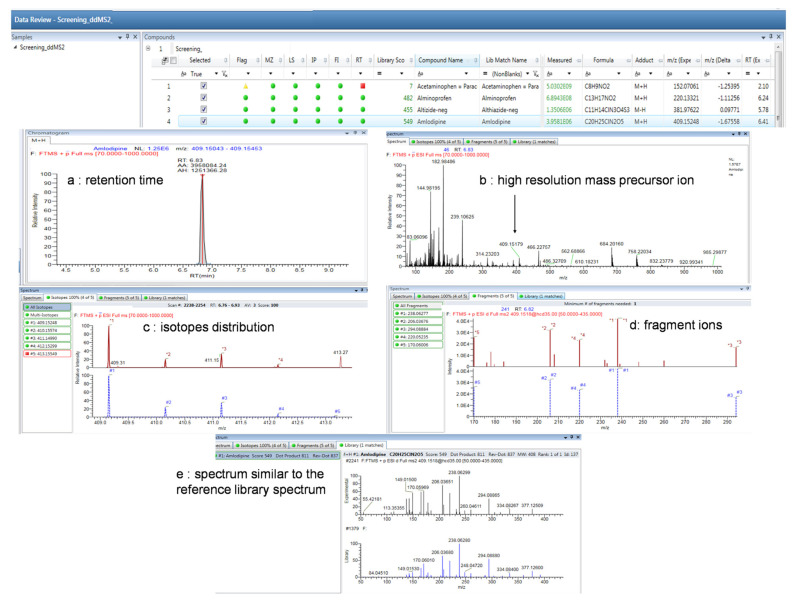
Example of amlodipine identification. Identification of amlodipine using 5 criteria: (**a**) retention time, (**b**) high-resolution mass precursor ion, (**c**) isotopic distributions, (**d**) fragment ions, and (**e**) HR–MS/MS spectrum similar to the reference library spectrum.

**Figure 4 pharmaceuticals-16-00076-f004:**
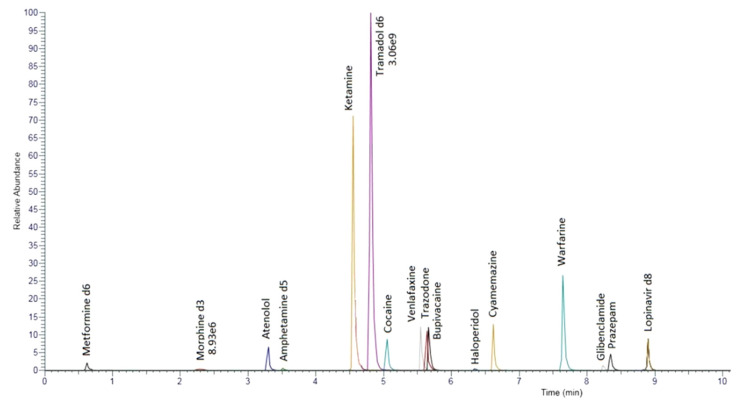
Representative reconstructed chromatogram of an internal plasma quality control containing 10 compounds and 6 internal standards.

**Table 1 pharmaceuticals-16-00076-t001:** Summary of the criteria used for the discrimination of isomeric compounds.

Compound Empirical Formula	RT (min)	*m*/*z*: Precursor *Relative Abundances*	*m*/*z*: Fragment Ion 1 *Relative Abundances*	*m*/*z*: Fragment Ion 2 *Relative Abundances*	*m*/*z*: Fragment Ion 3 *Relative Abundances*	*m*/*z*: Fragment Ion 4 *Relative Abundances*	*m*/*z*: Fragment Ion 5 *Relative Abundances*
O-Demethylvenlafaxine	4.13	264.19581	58.06605	246.1852	107.04956	133.06494	201.1275
C_16_H_25_NO_2_		*23.2*	*100*	*13.6*	*11.9*	*5.6*	*3.0*
Tramadol	4.42	264.19581	58.06595	182.9018	246.1843	265.1989	56.0503
C_16_H_25_NO_2_		*10.5*	*100*	*0.9*	*0.4*	*0.4*	*0.2*
MDEA	3.79	208.13321	163.07542	135.04408	133.06488	105.07027	72.08154
C_12_H_17_NO_2_		*15.2*	*100*	*33.4*	*31.4*	*31.3*	*9.5*
MBDB	4.01	208.13321	135.04413	177.09102	147.08046	72.08154	136.0475
C_12_H_17_NO_2_		*8.2*	*100*	*12.8*	*8.1*	*7.8*	*5.3*
Acepromazine	6.02	327.15256	86.09708	58.06606	254.06337	239.0765	222.0916
C_19_H_22_N_2_OS		*50.2*	*100*	*73.3*	*8.5*	*4.0*	*3.6*
Aceprométazine	5.94	327.15256	86.09696	240.04727	71.07362	239.07614	89.0602
C_19_H_22_N_2_OS		*14.8*	*100*	*20.3*	*6.8*	*5.8*	*2.5*
Morphine	1.27	286.14377	201.09116	229.08565	183.08067	185.05997	211.07568
C_17_H_19_NO_3_		*100*	*6.9*	*4.5*	*3.4*	*3.1*	*3.0*
Norcodeine	2.95	286.14377	268.13263	215.10689	225.09088	121.06505	243.10130
C_17_H_19_NO_3_		*100*	*10.7*	*6.5*	*5.4*	*5.3*	*4.7*
6-MAM	3.56	328.15433	211.07524	165.06987	193.06425	58.06528	183.08049
C_19_H_21_NO_4_		*100*	*13.1*	*7.5*	*7*	*6*	*4.5*
Naloxone	3.11	328.15433	310.1438	253.10934	268.13297	311.14734	269.1049
C_19_H_21_NO_4_		*75.4*	*100*	*22.3*	*19.5*	*11.6*	*10.3*
N-Demethylclobazam	6.42	287.05818	245.04753	210.07864	241.05254	269.0474	246.0511
C_15_H_11_ClN_2_0_2_		*29.2*	*100*	*13.7*	*12.4*	*4.8*	*2.7*
Oxazepam	6.40	287.05818	241.05254	269.04745	104.04984	231.06830	128.02629
C_15_H_11_ClN_2_0_2_		*40.4*	*100*	*40.9*	*17.6*	*15.3*	*8.7*

## Data Availability

Data are contained within the article or Appendix A.

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
