# Peer review of "Development and Validation of a Non-Targeted Screening Method for Most Psychoactive, Analgesic, Anaesthetic, Anti-Diabetic, Anti-Coagulant and Anti-Hypertensive Drugs in Human Whole Blood and Plasma Using High-Resolution Mass Spectrometry"

_pharmaceuticals, 2023, doi:10.3390/ph16010076_

Round 1

Reviewer 1 Report

1. Some terminologies are unfamiliar and need detailed decription. For example, it is not possible to determined whether the LOI referes to the detection limit or the quatification limit.

2. The analysis method is very important for all of the validation, but there is no information on the optimization or reference for the method used in this manuscript.

3. I still don't figure out ot the experimental conditions of matices. Such as spiked plasma, blank plasma, whole blood, etc.

Author Response

1/ Some terminologies are unfamiliar and need detailed decription. For example, it is not possible to determined whether the LOI referes to the detection limit or the quatification limit.

We thank the reviewer for your valuable commentaries. With their proposed corrections and our adjustments, we hope we will convince you of the value of our work

We thank you for your feedback and we propose to clarify the definition of LOI in paragraph 3.6.3 Limit of identification.

“The limit of identification was determined as the lowest concentration at which the substances were identified in all replicates based on the three following criteria [30] : 1) presence of high-resolution mass precursor ions, 2) presence of fragment ions and matching with library spectrum. Retention time and isotopic distribution were considered as minor criteria. These criteria for limit of identification were in accordance with the classification proposed by Broecker et al [17] and described in the paper of Helfer et al [30].”

  1. The analysis method is very important for all of the validation, but there is no information on the optimization or reference for the method used in this manuscript.

We totally agree with your relevant comment. We propose to add a paragraph as follows:

“2.1.1Chromatographic conditions and mass spectrometer parameters

Different chromatographic parameters were assayed and optimised to ensure good elution of the compounds with correct sensitivity. The first step of optimisation was the mass spectrometer parameters. A mixture of a pure solution of the 53 selected compounds in methanol was infused and then injected to optimise ionisation and mass spectrometer parameters. The second optimisation step was the chromatographic conditions with the choice of column, mobile phases and elution gradient. Acquity BEH HILIC (50 x 2.1 mm , 1.7 µm), Acquity HSS T3 (150 x 2.1 mm, 1.8 µm), Acquity UPLC® HSS C18 1.8 μm 2.1 × 150 mm (Waters Corp., Milford,  MA, USA) and Accucore Phenyl Hexyl UPLC 100 x 2.1 mm, 2.6 µm (ThermoFisher Scientific, San Jose, USA) columns were tested using the previous mixture. The chosen column was selected based on optimal shape of chromatographic peaks. Selected column was Accucore Phenyl Hexyl UPLC column, as it was for Helfer et al. [30,31]. Assayed mobile phases consisted of water + formic acid 0.1% (V/V) (mobile phase A) with or without ammonium acetate and acetonitrile + formic acid 0.1% (V/V) (mobile phase B). Mobile phases with ammonium acetate provided a better signal intensity than water + formic acid 0.1% (V/V). After these choices, the gradient was optimised.”

  1. I still don't figure out ot the experimental conditions of matices. Such as spiked plasma, blank plasma, whole blood, etc.

We totally agreed with this relevant comment. We propose to add this information in two paragraphs as follows.

“3.2. Preparation of stock solutions and working solutions

The internal standards were prepared as 1 g.L-1 stock solutions in acetonitrile. A working internal standard solution was then monthly prepared, containing 1 mg.L-1 lopinavir-D8, metformin-D6, tramadol-D6, trazodone-D6 and 0.1 mg.L-1 amphetamine-D5 and morphine-D3.

Other compounds were prepared as 1 g.L-1 stock solutions in methanol and stored at +4°C. Working solutions were prepared with appropriate serial cascade dilutions in methanol “

“3.6.3 Limit of identification

Whole blood and plasma samples were spiked with the 179 selected molecules at different concentrations according subtherapeutic, therapeutic and toxic concentrations, providing a large scale of concentrations [40,41] for each compounds. Therefore 10 µL of the working solutions were spiked into 100 µL of plasma or whole blood. Thereafter, the samples were prepared as described above.”

Reviewer 2 Report

This article describes the procedure for developing and validating a method for evaluation for large sample of toxicology drugs in human whole blood and plasma using high-resolution-mass-spectrometry.

In general, the article looks good, the main results of the study and the methodological part are described in detail.

The presented results are of great importance for toxicological screening and deserve publication in the journal Pharmaceuticals.

However, there are a number of points that raise doubts:

1) The choice of terminology used by the authors is unclear. For example, line 92: "The library initially provided by the supplier has been increased with molecules  whose identification by a screening analysis was of clinical interest." It seems to me more correct to use the word "compounds" rather than "molecules". The same thing is observed in lines 176-177.

2) Paragraph 2.1.2. The authors note that they have supplemented the library by adding 1488 new compounds. However, it is not at all clear where these compounds came from. Do I understand correctly that the authors purchased commercially available preparations of these 1488 compounds, recorded their chromatograms and MS spectra and entered them into the library?

3) The authors repeatedly refer to the tables and figures presented in the "Supplementary Material". However, this file was not submitted. Therefore, it is difficult to assess the correctness of the presented data.

4) Line 178: "The limits of identification in plasma were compared to the therapeutic concentrations reported in the literature." Authors should add the relevant references.

5) Paragraph 2.2.4. How can one explain the fact that Mean of library score ranged from such wide limits (from 45 to 973)?

6) Figures presented in the work are of poor quality and require improvements.

7) Figure 4 is missing.

8) In my opinion, paragraph 2.4 is not suitable for publication in the "Results and discussions". This is due to the fact that this section contains a significant proportion of the information that is usually presented in the "Introduction". I suggest that the authors remove this paragraph by transferring part of the information in the introduction, and part to the corresponding points of the experimental part. 

Author Response

This article describes the procedure for developing and validating a method for evaluation for large sample of toxicology drugs in human whole blood and plasma using high-resolution-mass-spectrometry.

In general, the article looks good, the main results of the study and the methodological part are described in detail.

The presented results are of great importance for toxicological screening and deserve publication in the journal Pharmaceuticals.

However, there are a number of points that raise doubts:

  • The choice of terminology used by the authors is unclear. For example, line 92: "The library initially provided by the supplier has been increased with molecules  whose identification by a screening analysis was of clinical interest." It seems to me more correct to use the word "compounds" rather than "molecules". The same thing is observed in lines 176-177.

We thank the reviewer for your valuable commentaries. With their proposed corrections and our adjustments, we hope we will convince you of the value of our work

We thank you for your suggestion. We have modified “molecules” into “compounds” in the text.

  • Paragraph 2.1.2. The authors note that they have supplemented the library by adding 1488 new compounds. However, it is not at all clear where these compounds came from. Do I understand correctly that the authors purchased commercially available preparations of these 1488 compounds, recorded their chromatograms and MS spectra and entered them into the library?

We totally agree and we propose to clarify the paragraph 2.1.2 :

“2.1.2 Library

The library initially provided by the supplier contained 1464 compounds. It has been further incremented with compounds whose identification by a screening analysis was of clinical interest. The following compounds were therefore added to the inclusion list after infusion of a pure solution:  baclofen, hydroxychloroquine, anticoagulants (apixaban, dabigatran, rivaroxaban, fluindione, tioclomarol and phenindione), rodenticides (difenacoum, diphenadione, chlorophacinone), antidiabetic such as sitagliptine and vildagliptine aztreonam, cefepime, cefotaxime, piperacillin, hydroxyalprazolam, isavuconazole, nordosulepine, norquetiapine, norsertraline, sulpiride, vortioxetine, oxomemazine. For these compounds, compound name, formula, polarity, high-resolution mass, retention time, isotopic distribution and HRMS spectrum were recorded to the library. Finally, the library consisted of 1489 compounds.”

  • The authors repeatedly refer to the tables and figures presented in the "Supplementary Material". However, this file was not submitted. Therefore, it is difficult to assess the correctness of the presented data.

We apologize for this error and this file is now uploaded in the new submission.

  • Line 178: "The limits of identification in plasma were compared to the therapeutic concentrations reported in the literature." Authors should add the relevant references.

We thank you for your feedback and we have incremented relevant references : “The LOD in plasma were compared to the therapeutic concentrations reported in the literature [40,41]”.

  • Paragraph 2.2.4. How can one explain the fact that Mean of library score ranged from such wide limits (from 45 to 973)?

We thank you for your answer. Library score was estimated using ThermoFisher algorithm as follow: the spectra of the different fragments of the compound and their intensities were compared between the observed spectra and the theoretical library spectra (dot product), then between the library spectra and the observed spectra (reverse dot product). As Helfer et al [30] and depending on compound, we fixed a minimum library score threshold to 20 A.U. Library score was not used as an identification criterion but as a quality indicator for the reproducibility of the spectra to assess the robustness of the method. In this assay, we do not evaluate the library score but its relative standard deviation by an inter-assay test.

  • Figures presented in the work are of poor quality and require improvements.

We thank you for your comment. We have reinserted all the figures in better quality.

  • Figure 4 is missing.

We apologize for this inconvenience and we have inserted figure 4 on page 14.

  • In my opinion, paragraph 2.4 is not suitable for publication in the "Results and discussions". This is due to the fact that this section contains a significant proportion of the information that is usually presented in the "Introduction". I suggest that the authors remove this paragraph by transferring part of the information in the introduction, and part to the corresponding points of the experimental part.

We have considered your advice. Paragraph 2.4 has been removed and the information has been transferred to the introduction and to the different parts of paragraph 2 Results and discussion.

Round 2

Reviewer 1 Report

Well revised.